# Convex Tensor Decomposition via Structured Schatten Norm Regularization

**Ryota Tomioka**
Toyota Technological Institute at Chicago
Chicago, IL 60637
tomioka@ttic.edu

**Taiji Suzuki**
Department of Mathematical
and Computing Sciences
Tokyo Institute of Technology
Tokyo 152-8552, Japan
s-taiji@is.titech.ac.jp

## Abstract

We study a new class of structured Schatten norms for tensors that includes two recently proposed norms ("overlapped" and "latent") for convex-optimization-based tensor decomposition. We analyze the performance of "latent" approach for tensor decomposition, which was empirically found to perform better than the "overlapped" approach in some settings. We show theoretically that this is indeed the case. In particular, when the unknown true tensor is low-rank in a specific unknown mode, this approach performs as well as knowing the mode with the smallest rank. Along the way, we show a novel duality result for structured Schatten norms, which is also interesting in the general context of structured sparsity. We confirm through numerical simulations that our theory can precisely predict the scaling behaviour of the mean squared error.

## 1 Introduction

Decomposition of tensors [10, 14] (or multi-way arrays) into low-rank components arises naturally in many real world data analysis problems. For example, in neuroimaging, spatio-temporal patterns of neural activities that are related to certain experimental conditions or subjects can be found by computing the tensor decomposition of the data tensor, which can be of size channels $\times$ time-points $\times$ subjects $\times$ conditions [18]. More generally, any multivariate spatio-temporal data (e.g., environmental monitoring) can be regarded as a tensor. If some of the observations are missing, low-rank modeling enables the imputation of missing values. Tensor modelling may also be valuable for collaborative filtering with temporal or contextual dimension.

Conventionally, tensor decomposition has been tackled through non-convex optimization problems, using alternate least squares or higher-order orthogonal iteration [6]. Compared to its empirical success, little has been theoretically understood about the performance of tensor decomposition algorithms. De Lathauwer et al. [5] showed an approximation bound for a truncated higher-order SVD (also known as the Tucker decomposition). Nevertheless the generalization performance of these approaches has been widely open. Moreover, the model selection problem can be highly challenging, especially for the Tucker model [5, 27], because we need to specify the rank $r_k$ for each mode (here a mode refers to one dimensionality of a tensor); that is, we have $K$ hyper-parameters to choose for a $K$-way tensor, which is challenging even for $K = 3$.

Recently a convex-optimization-based approach for tensor decomposition has been proposed by several authors [9, 15, 23, 25], and its performance has been analyzed in [26].

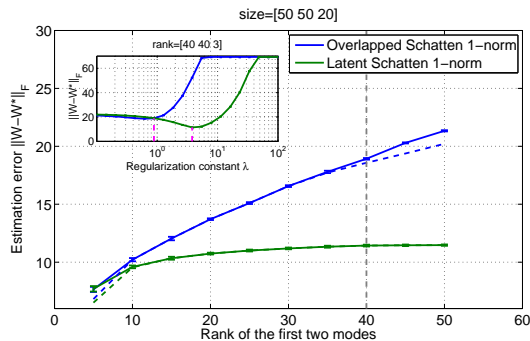

Figure 1: Estimation of a low-rank $50\times50\times20$ tensor of rank $r \times r \times 3$ from noisy measurements. The noise standard deviation is $\sigma = 0.1$. The estimation errors of two convex optimization based methods are plotted against the rank $r$ of the first two modes. The solid lines show the error at the fixed regularization constant $\lambda$, which is 0.89 for the overlapped approach and 3.79 for the latent approach (see also Figure 2). The dashed lines show the minimum error over candidates of the regularization constant $\lambda$ from 0.1 to 100. In the inset, the errors of the two approaches are plotted against the regularization constant $\lambda$ for rank $r = 40$ (marked with gray dashed vertical line in the outset). The two values (0.89 and 3.79) are marked with vertical dashed lines. Note that both approaches need no knowledge of the true rank; the rank is automatically learned.

The basic idea behind their convex approach, which we call *overlapped approach*, is to unfold[1] a tensor into matrices along different modes and penalize the unfolded matrices to be *simultaneously low-rank* based on the Schatten 1-norm, which is also known as the trace norm and nuclear norm [7, 22, 24]. This approach does not require the rank of the decomposition to be specified beforehand, and due to the low-rank inducing property of the Schatten 1-norm, the rank of the decomposition is *automatically* determined.

However, it has been noticed that the above overlapped approach has a limitation that it performs poorly for a tensor that is only low-rank in a certain mode. The authors of [25] proposed an alternative approach, which we call *latent approach*, that decomposes a given tensor into a mixture of tensors that each are low-rank in a specific mode. Figure 1 demonstrates that the latent approach is preferable to the overlapped approach when the underlying tensor is almost full rank in all but one mode. However, so far no theoretical analysis has been presented to support such an empirical success.

In this paper, we rigorously study the performance of the latent approach and show that the mean squared error of the latent approach scales no greater than the minimum mode-$k$ rank of the underlying true tensor, which clearly explains why the latent approach performs better than the overlapped approach in Figure 1.

Along the way, we show a novel *duality* between the two types of norms employed in the above two approaches, namely the overlapped Schatten norm and the latent Schatten norm. This result is closely related and generalize the results in structured sparsity literature [2, 13, 17, 21]. In fact, the *(plain) overlapped group lasso* constrains the weights to be simultaneously group sparse over overlapping groups. The *latent group lasso* predicts with a mixture of group sparse weights [see also 1, 3, 12]. These approaches clearly correspond to the two variations of tensor decomposition algorithms we discussed above.

Finally we empirically compare the overlapped approach and latent approach and show that even when the unknown tensor is simultaneously low-rank, which is a favorable situation for the overlapped approach, the latent approach performs better in many cases. Thus we provide both theoretical and empirical evidence that for noisy tensor decomposition, the latent approach is preferable to the overlapped approach. Our result is complementary to the previous study [25, 26], which mainly focused on the noise-less tensor completion setting.

This paper is structured as follows. In Section 2, we provide basic definitions of the two variations of structured Schatten norms, namely the overlapped/latent Schatten norms, and discuss their properties, especially the *duality* between them. Section 3 presents our main theoretical contributions; we establish the consistency of the latent approach, and we analyze the denoising performance of the latent approach. In Section 4, we empirically confirm the scaling predicted by our theory. Finally, Section 5 concludes the paper. Most of the proofs are presented in the supplementary material.

## 2 Structured Schatten norms for tensors

In this section, we define the overlapped Schatten norm and the latent Schatten norm and discuss their basic properties.

First we need some basic definitions.

Let $\mathcal{W} \in \mathbb{R}^{n_1 \times \cdots n_K}$ be a $K$-way tensor. We denote the total number of entries in $\mathcal{W}$ by $N = \prod_{k=1}^{K} n_k$. The dot product between two tensors $\mathcal{W}$ and $\mathcal{X}$ is defined as $\langle \mathcal{W}, \mathcal{X} \rangle = \mathrm{vec}(\mathcal{W})^\top \mathrm{vec}(\mathcal{X})$; i.e., the dot product as vectors in $\mathbb{R}^N$. The Frobenius norm of a tensor is defined as $\|\mathcal{W}\|_F = \sqrt{\langle \mathcal{W}, \mathcal{W} \rangle}$. Each dimensionality of a tensor is called a *mode*. The mode $k$ *unfolding* $\boldsymbol{W}_{(k)} \in \mathbb{R}^{n_k \times N/n_k}$ is a matrix that is obtained by concatenating the mode-$k$ fibers along columns; here a mode-$k$ fiber is an $n_k$ dimensional vector obtained by fixing all the indices but the $k$th index of $\mathcal{W}$. The mode-$k$ rank $\underline{r}_k$ of $\mathcal{W}$ is the rank of the mode-$k$ unfolding $\boldsymbol{W}_{(k)}$. We say that a tensor $\mathcal{W}$ has multilinear rank $(\underline{r}_1, \ldots, \underline{r}_K)$ if the mode-$k$ rank is $\underline{r}_k$ for $k = 1, \ldots, K$ [14]. The mode $k$ folding is the inverse of the unfolding operation.

### 2.1 Overlapped Schatten norms

The low-rank inducing norm studied in [9, 15, 23, 25], which we call overlapped Schatten 1-norm, can be written as follows:

$$\|\mathcal{W}\|_{\underline{S_1/1}} = \sum\nolimits_{k=1}^{K} \|\boldsymbol{W}_{(k)}\|_{S_1}. \tag{1}$$

In this paper, we consider the following more general *overlapped $S_p/q$-norm*, which includes the Schatten 1-norm as the special case $(p, q) = (1, 1)$. The overlapped $S_p/q$-norm is written as follows:

$$\|\mathcal{W}\|_{\underline{S_p/q}} = \left( \sum\nolimits_{k=1}^{K} \|\boldsymbol{W}_{(k)}\|_{S_p}^q \right)^{1/q}, \tag{2}$$

where $1 \le p, q \le \infty$; here

$$\|\boldsymbol{W}\|_{S_p} = \left( \sum\nolimits_{j=1}^{r} \sigma_j^p(\boldsymbol{W}) \right)^{1/p}$$

is the Schatten $p$-norm for matrices, where $\sigma_j(\boldsymbol{W})$ is the $j$th largest singular value of $\boldsymbol{W}$.

When used as a regularizer, the overlapped Schatten 1-norm penalizes all modes of $\mathcal{W}$ to be jointly low-rank. It is related to the overlapped group regularization [see 13, 16] in a sense that the same object $\mathcal{W}$ appears repeatedly in the norm.

The following inequality relates the overlapped Schatten 1-norm with the Frobenius norm, which was a key step in the analysis of [26]:

$$\|\mathcal{W}\|_{\underline{S_1/1}} \le \sum_{k=1}^{K} \sqrt{\underline{r}_k} \|\mathcal{W}\|_F, \tag{3}$$

where $\underline{r}_k$ is the mode-$k$ rank of $\mathcal{W}$.

Now we are interested in the dual norm of the overlapped $S_p/q$-norm, because deriving the dual norm is a key step in solving the minimization problem that involves the norm (2) [see 16], as well as computing various complexity measures, such as, Rademacher complexity [8] and Gaussian width [4]. It turns out that the dual norm of the overlapped $S_p/q$-norm is the *latent $S_{p^*}/q^*$-norm* as shown in the following lemma (proof is presented in Appendix A).

**Lemma 1.** *The dual norm of the overlapped $S_p/q$-norm is the latent $S_{p^*}/q^*$-norm, where $1/p + 1/p^* = 1$ and $1/q + 1/q^* = 1$, which is defined as follows:*

$$\||\mathcal{X}\||_{\overline{S_{p^*}/q^*}} = \inf_{\left(\mathcal{X}^{(1)}+\cdots+\mathcal{X}^{(K)}\right)=\mathcal{X}} \left(\sum_{k=1}^{K} \|\boldsymbol{X}_{(k)}^{(k)}\|_{S_{p^*}}^{q^*}\right)^{1/q^*}. \tag{4}$$

*Here the infimum is taken over the $K$-tuple of tensors $\mathcal{X}^{(1)}, \ldots, \mathcal{X}^{(K)}$ that sums to $\mathcal{X}$.*

In the supplementary material, we show a slightly more general version of the above lemma that naturally generalizes the duality between overlapped/latent group sparsity norms [1, 12, 17, 21]; see Section A. Note that when the groups have no overlap, the overlapped/latent group sparsity norms become identical, and the duality is the ordinary duality between the group $S_p/q$-norms and the group $S_{p^*}/q^*$-norms.

## 2.2 Latent Schatten norms

The latent approach for tensor decomposition [25] solves the following minimization problem

$$\underset{\mathcal{W}^{(1)}, \ldots, \mathcal{W}^{(K)}}{\text{minimize}} \quad L(\mathcal{W}^{(1)} + \cdots + \mathcal{W}^{(K)}) + \lambda \sum_{k=1}^{K} \|\boldsymbol{W}_{(k)}^{(k)}\|_{S_1}, \tag{5}$$

where $L$ is a loss function, $\lambda$ is a regularization constant, and $\boldsymbol{W}_{(k)}^{(k)}$ is the mode-$k$ unfolding of $\mathcal{W}^{(k)}$. Intuitively speaking, the latent approach for tensor decomposition predicts with a mixture of $K$ tensors that each are regularized to be low-rank in a specific mode.

Now, since the loss term in the minimization problem (5) only depends on the sum of the tensors $\mathcal{W}^{(1)}, \ldots, \mathcal{W}^{(K)}$, minimization problem (5) is equivalent to the following minimization problem

$$\underset{\mathcal{W}}{\text{minimize}} \quad L(\mathcal{W}) + \lambda \||\mathcal{W}\||_{\overline{S_1/1}}.$$

In other words, we have identified the structured Schatten norm employed in the latent approach as the latent $S_1/1$-norm (or latent Schatten 1-norm for short), which can be written as follows:

$$\||\mathcal{W}\||_{\overline{S_1/1}} = \inf_{\left(\mathcal{W}^{(1)}+\cdots+\mathcal{W}^{(K)}\right)=\mathcal{W}} \sum_{k=1}^{K} \|\boldsymbol{W}_{(k)}^{(k)}\|_{S_1}. \tag{6}$$

According to Lemma 1, the dual norm of the latent $S_1/1$-norm is the overlapped $S_\infty/\infty$-norm

$$\||\mathcal{X}\||_{\underline{S_\infty/\infty}} = \max_k \|\boldsymbol{X}_{(k)}\|_{S_\infty}, \tag{7}$$

where $\|\cdot\|_{S_\infty}$ is the spectral norm.

The following lemma is similar to inequality (3) and is a key in our analysis (proof is presented in Appendix B).

**Lemma 2.**

$$\||\mathcal{W}\||_{\overline{S_1/1}} \leq \left(\min_k \sqrt{\underline{r_k}}\right) \||\mathcal{W}\||_F,$$

*where $\underline{r_k}$ is the mode-$k$ rank of $\mathcal{W}$.*

Compared to inequality (3), the latent Schatten 1-norm is bounded by the *minimal* square root of the ranks instead of the sum. This is the fundamental reason why the latent approach performs betters than the overlapped approach as in Figure 1.

## 3 Main theoretical results

In this section, combining the duality we presented in the previous section with the techniques from Agarwal et al. [1], we study the generalization performance of the latent approach for tensor decomposition in the context of recovering an unknown tensor $\mathcal{W}^*$ from noisy measurements. This is the setting of the experiment in Figure 1. We first prove a generic consistency statement that does not take the low-rank-ness of the truth into account. Next we show that a tighter bound that takes the low-rank-ness into account can be obtained with some incoherence assumption. Finally, we discuss the difference between overlapped approach and latent approach and provide an explanation for the empirically observed superior performance of the latent approach in Figure 1.

## 3.1 Consistency

Let $\mathcal{W}^*$ be the underlying true tensor and the noisy version $\mathcal{Y}$ is obtained as follows:

$$\mathcal{Y} = \mathcal{W}^* + \mathcal{E},$$

where $\mathcal{E} \in \mathbb{R}^{n_1 \times \cdots \times n_K}$ is the noise tensor.

A consistency statement can be obtained as follows (proof is presented in Appendix C):

**Theorem 1.** *Assume that the regularization constant $\lambda$ satisfies $\lambda \geq \left\|\left\|\mathcal{E}\right\|\right\|_{S_\infty/\infty}$ (overlapped $S_\infty/\infty$ norm of the noise), then the estimator defined by $\hat{\mathcal{W}} = \mathrm{argmin}_{\mathcal{W}} \left( \frac{1}{2}\left\|\mathcal{Y} - \mathcal{W}\right\|_F^2 + \lambda \left\|\left\|\mathcal{W}\right\|\right\|_{\overline{S_1/1}} \right),$ satisfies the inequality*

$$\left\|\left\|\hat{\mathcal{W}} - \mathcal{W}^*\right\|\right\|_F \leq 2\lambda \sqrt{\min_k n_k}. \tag{8}$$

In particular when the noise goes to zero $\mathcal{E} \to 0$, the right hand side of inequality (8) shrinks to zero.

## 3.2 Deterministic bound

The consistency statement in the previous section only deals with the sum $\hat{\mathcal{W}} = \sum_{k=1}^{K} \hat{\mathcal{W}}^{(k)}$ and the statement does not take into account the low-rank-ness of the truth. In this section, we establish a tighter statement that bounds the errors of individual terms $\hat{\mathcal{W}}^{(k)}$.

To this end, we need some additional assumptions. First, we assume that the unknown tensor $\mathcal{W}^*$ is a mixture of $K$ tensors that each are low-rank in a certain mode and we have a noisy observation $\mathcal{Y}$ as follows:

$$\mathcal{Y} = \mathcal{W}^* + \mathcal{E} = \sum_{k=1}^{K} \mathcal{W}^{*(k)} + \mathcal{E}, \tag{9}$$

where $\bar{r}_k = \mathrm{rank}(\boldsymbol{W}_{(k)}^{(k)})$ is the mode-$k$ rank of the $k$th component $\mathcal{W}^{*(k)}$; note that this does not equal the mode-$k$ rank $\underline{r}_k$ of $\mathcal{W}^*$ in general.

Second, we assume that the spectral norm of the mode-$k$ unfolding of the $l$th component is bounded by a constant $\alpha$ for all $k \neq l$ as follows:

$$\|\boldsymbol{W}_{(k)}^{*(l)}\|_{S_\infty} \leq \alpha \quad (\forall l \neq k, k, l = 1, \ldots, K). \tag{10}$$

Note that such an additional incoherence assumption has also been used in [1, 3, 11].

We employ the following optimization problem to recover the unknown tensor $\mathcal{W}^*$:

$$\hat{\mathcal{W}} = \mathrm{argmin}_{\mathcal{W}} \left( \frac{1}{2}\left\|\mathcal{Y} - \mathcal{W}\right\|_F^2 + \lambda \left\|\left\|\mathcal{W}\right\|\right\|_{\overline{S_1/1}} \quad \text{s.t. } \mathcal{W} = \sum_{k=1}^{K} \mathcal{W}^{(k)}, \ \|\boldsymbol{W}_{(k)}^{(l)}\|_{S_\infty} \leq \alpha, \quad \forall l \neq k \right), \tag{11}$$

where $\lambda > 0$ is a regularization constant. Notice that we have introduced additional spectral norm constraints to control the correlation between the components; see also [1].

Our deterministic performance bound can be stated as follows (proof is presented in Appendix D):

**Theorem 2.** *Let $\hat{\mathcal{W}}^{(k)}$ be an optimal decomposition of $\hat{\mathcal{W}}$ induced by the latent Schatten 1-norm (6). Assume that the regularization constant $\lambda$ satisfies $\lambda \geq 2\left\|\left\|\mathcal{E}\right\|\right\|_{S_\infty/\infty} + \alpha(K-1)$. Then there is a universal constant $c$ such that, any solution $\hat{\mathcal{W}}$ of the minimization problem (11) satisfies the following deterministic bound:*

$$\sum_{k=1}^{K} \left\|\left\|\hat{\mathcal{W}}^{(k)} - \mathcal{W}^{*(k)}\right\|\right\|_F^2 \leq c\lambda^2 \sum_{k=1}^{K} \bar{r}_k. \tag{12}$$

*Moreover, the overall error can be bounded in terms of the multilinear rank of $\mathcal{W}^*$ as follows:*

$$\left\|\left\|\hat{\mathcal{W}} - \mathcal{W}^*\right\|\right\|_F^2 \leq c\lambda^2 \min_k \underline{r}_k. \tag{13}$$

Note that in order to get inequality (13), we exploit the arbitrariness of the decomposition $\mathcal{W}^* = \sum_{k=1}^{K} \mathcal{W}^{*(k)}$ to replace the sum over the ranks with the minimal *mode-k rank*. This is possible because a *singleton decomposition*, i.e., $\mathcal{W}^{*(k)} = \mathcal{W}^*$ and $\mathcal{W}^{*(k')} = 0$ for $k' \neq k$, is allowed for any $k$.

Comparing two inequalities (8) and (13), we see that there are two regimes. When the noise is small, (8) is tighter. On the other hand, when the noise is larger and/or $\min_k \underline{r}_k \ll \min_k n_k$, (13) is tighter.

### 3.3 Gaussian noise

When the elements of the noise tensor $\mathcal{E}$ are Gaussian, we obtain the following theorem.

**Theorem 3.** *Assume that the elements of the noise tensor $\mathcal{E}$ are independent zero-mean Gaussian random variables with variance $\sigma^2$. In addition, assume without loss of generality that the dimensionalities of $\mathcal{W}^*$ are sorted in the descending order, i.e., $n_1 \geq \cdots \geq n_K$. Then there is a universal constant $c$ such that, with probability at least $1 - \delta$, any solution of the minimization problem (11) with regularization constant $\lambda = 2\sigma(\sqrt{N/n_K} + \sqrt{n_1} + \sqrt{2\log(K/\delta)}) + \alpha(K-1)$ satisfies*

$$\frac{1}{N} \sum_{k=1}^{K} \left\| \hat{\mathcal{W}}^{(k)} - \mathcal{W}^{*(k)} \right\|_F^2 \leq cF\sigma^2 \frac{\sum_{k=1}^{K} \bar{r}_k}{n_K}, \tag{14}$$

*where $F = \left( \left(1 + \sqrt{\frac{n_1 n_K}{N}}\right) + \left(\sqrt{2\log(K/\delta)} + \frac{\alpha(K-1)}{2\sigma}\right) \sqrt{\frac{n_K}{N}} \right)^2$ is a factor that mildly depends on the dimensionalities and the constant $\alpha$ in (10).*

Note that the theoretically optimal choice of regularization constant $\lambda$ is independent of the ranks of the truth $\mathcal{W}^*$ or its factors in (9), which are unknown in practice.

Again we can obtain a bound corresponding to the minimum rank singleton decomposition as in inequality (13) as follows:

$$\frac{1}{N} \left\| \hat{\mathcal{W}} - \mathcal{W}^* \right\|_F^2 \leq cF\sigma^2 \frac{\min_k \underline{r}_k}{n_K}, \tag{15}$$

where $F$ is the same factor as in Theorem 3.

### 3.4 Comparison with the overlapped approach

Inequality (15) explains the superior performance of the latent approach for tensor decomposition in Figure 1. The inequality obtained in [26] for the overlapped approach that uses overlapped Schatten 1-norm (1) can be stated as follows:

$$\frac{1}{N} \left\| \hat{\mathcal{W}} - \mathcal{W}^* \right\|_F^2 \leq c'\sigma^2 \left( \frac{1}{K} \sum_{k=1}^{K} \sqrt{\frac{1}{n_k}} \right)^2 \left( \frac{1}{K} \sum_{k=1}^{K} \sqrt{\underline{r}_k} \right)^2. \tag{16}$$

Comparing inequalities (15) and (16), we notice that the complexity of the overlapped approach depends on the average (square root) of the mode-$k$ ranks $\underline{r}_1, \ldots, \underline{r}_K$, whereas that of the latent approach only grows linearly against the *minimum* mode-$k$ rank. Interestingly, the latent approach performs *as if it knows the mode with the minimum rank*, although such information is not given.

Recently, Mu et al. [19] proved a lower bound of the number of measurements for solving linear inverse problem via the overlapped approach. Although the setting is different, the lower bound depends on the minimum mode-$k$ rank, which agrees with the complexity of the latent approach.

## 4 Numerical results

In this section, we numerically confirm the theoretically obtained scaling behavior.

The goal of this experiment is to recover the true low rank tensor $\mathcal{W}^*$ from a noisy observation $\mathcal{Y}$. We randomly generated the true low rank tensors $\mathcal{W}^*$ of size $50 \times 50 \times 20$ or $80 \times 80 \times 40$ with various mode-$k$ ranks $(\underline{r}_1, \underline{r}_2, \underline{r}_3)$. A low-rank tensor is generated by first randomly drawing the

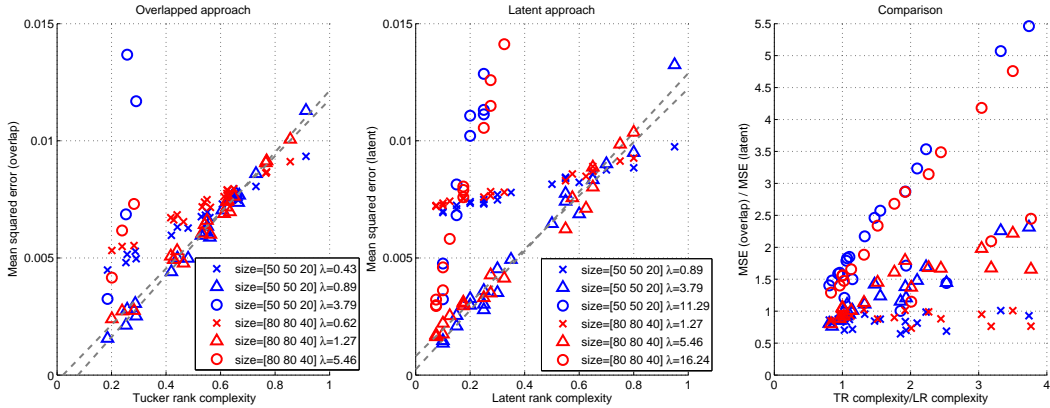

Figure 2: Performance of the overlapped approach and latent approach for tensor decomposition are shown against their theoretically predicted complexity measures (see Eqs. (17) and (18)). The right panel shows the improvement of the latent approach from the overlapped approach against the ratio of their complexity measures.

$\underline{r}_1 \times \underline{r}_2 \times \underline{r}_3$ core tensor from the standard normal distribution and multiplying an orthogonal factor matrix drawn uniformly to its each mode. The observation tensor $\mathcal{Y}$ is obtained by adding Gaussian noise with standard deviation $\sigma = 0.1$. There is no missing entries in this experiment.

For each observation $\mathcal{Y}$, we computed tensor decompositions using the overlapped approach and the latent approach (11). For the optimization, we used the algorithms[2] based on alternating direction method of multipliers described in Tomioka et al. [25]. We computed the solutions for 20 candidate regularization constants ranging from 0.1 to 100 and report the results for three representative values for each method.

We measured the quality of the solutions obtained by the two approaches by the mean squared error (MSE) $\left\| \hat{\mathcal{W}} - \mathcal{W}^* \right\|_F^2 / N$. In order to make our theoretical predictions more concrete, we define the quantities in the right hand side of the bounds (16) and (14) as *Tucker rank (TR) complexity* and *Latent rank (LR) complexity*, respectively, as follows:

$$\text{TR complexity} = \left( \frac{1}{K} \sum_{k=1}^{K} \sqrt{\frac{1}{n_k}} \right)^2 \left( \frac{1}{K} \sum_{k=1}^{K} \sqrt{\underline{r}_k} \right)^2, \tag{17}$$

$$\text{LR complexity} = \frac{\sum_{k=1}^{K} \bar{r}_k}{n_K}, \tag{18}$$

where without loss of generality we assume $n_1 \geq \cdots \geq n_K$. We have ignored terms like $\sqrt{n_k/N}$ because they are negligible for $n_k \approx 50$ and $N \approx 50,000$. The TR complexity is equivalent to the *normalized rank* in [26]. Note that the TR complexity (17) is defined in terms of the multilinear rank $(\underline{r}_1, \ldots, \underline{r}_K)$ of the truth $\mathcal{W}^*$, whereas the LR complexity (18) is defined in terms of the ranks of the latent factors $(\bar{r}_1, \ldots, \bar{r}_K)$ in (9). In order to find a decomposition that minimizes the right hand side of (18), we ran the latent approach to the true tensor $\mathcal{W}^*$ without noise, and took the minimum of the sum of ranks found by the run and $\min_k \underline{r}_k$, i.e., the minimal mode-$k$ rank (because a singleton solution is also allowed). The whole procedure is repeated 10 times and averaged.

Figure 2 shows the results of the experiment. The left panel shows the MSE of the overlapped approach against the TR complexity (17). The middle panel shows the MSE of the latent approach against the LR complexity (18). The right panel shows the improvement (i.e., MSE of the overlap approach over that of the latent approach) against the ratio of the respective complexity measures.

First, from the left panel, we can confirm that as predicted by [26], the MSE of the overlapped approach scales linearly against the TR complexity (17) for each value of the regularization constant.

From the central panel, we can clearly see that the MSE of the latent approach scales linearly against the LR complexity (18) as predicted by Theorem 3. The series with $\triangle$ ($\lambda = 3.79$ for $50 \times 50 \times 20$,

$\lambda = 5.46$ for $80 \times 80 \times 40$) is mostly below other series, which means that the optimal choice of the regularization constant is independent of the rank of the true tensor and only depends on the size; this agrees with the condition on $\lambda$ in Theorem 3. Since the blue series and red series with the same markers lie on top of each other (especially the series with $\triangle$ for which the optimal regularization constant is chosen), we can see that our theory predicts not only the scaling against the latent ranks but also that against the size of the tensor correctly. Note that the regularization constants are scaled by roughly 1.6 to account for the difference in the dimensionality.

The right panel reveals that in many cases the latent approach performs better than the overlapped approach, i.e., MSE (overlap)/ MSE (latent) greater than one. Moreover, we can see that the success of the latent approach relative to the overlapped approach is correlated with high TR complexity to LR complexity ratio. Indeed, we found that an optimal decomposition of the true tensor $\mathcal{W}^*$ was typically a singleton decomposition corresponding to the smallest tucker rank (see Section 3.2). Note that the two approaches perform almost identically when they are under-regularized (crosses).

The improvements here are milder than that in Figure 1. This is because most of the randomly generated low-rank tensors were simultaneously low-rank to some degree. It is encouraging that the latent approach perform at least as well as the overlapped approach in such situations as well.

## 5    Conclusion

In this paper, we have presented a framework for structured Schatten norms. The current framework includes both the overlapped Schatten 1-norm and latent Schatten 1-norm recently proposed in the context of convex-optimization-based tensor decomposition [9, 15, 23, 25], and connects these studies to the broader studies on structured sparsity [2, 13, 17, 21]. Moreover, we have shown a *duality* that holds between the two types of norms.

Furthermore, we have rigorously studied the performance of the latent approach for tensor decomposition. We have shown the consistency of the latent Schatten 1-norm minimization. Next, we have analyzed the denoising performance of the latent approach and shown that the error of the latent approach is upper bounded by the *minimal* mode-$k$ rank, which contrasts sharply against the average (square root) dependency of the overlapped approach analyzed in [26]. This explains the empirically observed superior performance of the latent approach compared to the overlapped approach. The most difficult case for the overlapped approach is when the unknown tensor is only low-rank in one mode as in Figure 1.

We have also confirmed through numerical simulations that our analysis precisely predicts the scaling of the mean squared error as a function of the dimensionalities and the sum of ranks of the factors of the unknown tensor, which is dominated by the minimal mode-$k$ rank. Unlike mode-$k$ ranks, the ranks of the factors are not easy to compute. However, note that the theoretically optimal choice of the regularization constant does not depend on these quantities.

Thus, we have theoretically and empirically shown that for noisy tensor decomposition, the latent approach is more likely to perform better than the overlapped approach. Analyzing the performance of the latent approach for tensor completion would be an important future work.

The structured Schatten norms proposed in this paper include norms for tensors that are not employed in practice yet. Therefore, it would be interesting to explore various extensions, such as, using the overlapped $S_1/\infty$-norm instead of the $S_1/1$-norm or a *non-sparse* tensor decomposition.

**Acknowledgment**: This work was carried out while both authors were at The University of Tokyo. This work was partially supported by JSPS KAKENHI 25870192 and 25730013, and the Aihara Project, the FIRST program from JSPS, initiated by CSTP.

## Footnotes

[1]For a $K$-way tensor, there are $K$ ways to unfold a tensor into a matrix. See Section 2.

[2]The solver is available online: https://github.com/ryotat/tensor.

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
