[Supplementary Material]

# Supplementary material for "Convex Tensor Decomposition via Structured Schatten Norms"

## A  Proof of Lemma 1

*Proof.* This follows from the following lemma.

**Lemma 3.** *Let $\| \cdot \|_\star$ be a norm and $\boldsymbol{\Phi}$ be a linear operator from $\mathbb{R}^N$ to $\mathbb{R}^M$. Assume that the right kernel of $\boldsymbol{\Phi}$ is empty. Then*

1. $\left\|\mathcal{W}\right\|_{\underline{\star}(\boldsymbol{\Phi})} := \|\boldsymbol{\Phi}(\mathcal{W})\|_\star$ *is a norm.*

2. $\left\|\mathcal{W}\right\|_{\overline{\star}(\boldsymbol{\Phi})} := \inf_{\boldsymbol{z} \in \mathbb{R}^M} \|\boldsymbol{z}\|_\star \quad \text{s.t.} \quad \boldsymbol{\Phi}^\top(\boldsymbol{z}) = \mathcal{W}$ *is also a norm.*

3. $\left\|\cdot\right\|_{\underline{\star}(\boldsymbol{\Phi})}$ *and* $\left\|\cdot\right\|_{\overline{\star^*}(\boldsymbol{\Phi})}$ *are dual to each other, where* $\| \cdot \|_{\star^*}$ *is the dual norm of* $\| \cdot \|_\star$.

In fact, let $M = KN$ and let $\boldsymbol{\Phi}(\mathcal{W}) = [\text{vec}(\boldsymbol{W}_{(1)}); ...; \text{vec}(\boldsymbol{W}_{(K)})]$, i.e., the column-wise concatenation of unfoldings of $\mathcal{W}$ vectorized, and define the $\star$-norm $\| \cdot \|_\star$ as follows:

$$\|\boldsymbol{z}\|_\star = \left(\sum\nolimits_{k=1}^K \|\boldsymbol{Z}_{(k)}^{(k)}\|_{S_p}^q\right)^{1/q}, \tag{19}$$

where $\boldsymbol{Z}_{(k)}^{(k)}$ denotes the inverse vectorization of an $N$ dimensional sub-vector $\boldsymbol{z}_{(k-1)N+1:kN}$ of $\boldsymbol{z}$ into an $n_k \times N/n_k$ matrix. Now the dual norm of $\star$-norm (19) can be written as follows:

$$\|\boldsymbol{z}\|_{\star^*} = \left(\sum\nolimits_{k=1}^K \|\boldsymbol{Z}_{(k)}^{(k)}\|_{S_{p^*}}^{q^*}\right)^{1/q^*}.$$

Furthermore, the constraint $\boldsymbol{\Phi}^\top(\boldsymbol{z}) = \mathcal{X}$ can be easily rewritten as follows:

$$\sum\nolimits_{k=1}^K \mathcal{Z}^{(k)} = \mathcal{X},$$

where $\mathcal{Z}^{(k)}$ is the mode-$k$ folding (inverse unfolding) of $\boldsymbol{Z}_{(k)}^{(k)}$. By replacing $\mathcal{Z}^{(k)}$ by $\mathcal{X}^{(k)}$ for $k = 1, \ldots, K$, we have Lemma 1.

Now we prove Lemma 3. The first and the second parts are straightforward to show. In order to prove the third part, let's start from the definition of a dual norm

$$\|\mathcal{X}\|_{\underline{\star}(\boldsymbol{\Phi})^*} = \sup \langle \mathcal{W}, \mathcal{X} \rangle \quad \text{s.t.} \quad \left\|\mathcal{W}\right\|_{\underline{\star}(\boldsymbol{\Phi})} \leq 1. \tag{20}$$

This is a constrained maximization problem. Since the above maximization problem satisfies Slater's condition, the strong duality holds. Thus, we only need to show that its dual problem agrees with the definition of $\left\|\cdot\right\|_{\overline{\star^*}(\boldsymbol{\Phi})}$. Due to Fenchel's duality theorem, we have

$$\sup_{\mathcal{W}} \left(\langle \mathcal{W}, \mathcal{X} \rangle - \delta\left(\|\boldsymbol{\Phi}(\mathcal{W})\|_\star \leq 1\right)\right) = \inf_{\boldsymbol{z}} \left(\delta(-\boldsymbol{\Phi}^\top(\boldsymbol{z}) + \mathcal{X} = 0) + \|\boldsymbol{z}\|_{\star^*}\right),$$

where $\delta(C)$ is the indicator function of condition $C$ (0 if $C$ is true, and $\infty$ otherwise). Now we can identify the left-hand side as the definition of dual norm (20) and the right-hand side as $\left\|\mathcal{X}\right\|_{\overline{\star^*}(\boldsymbol{\Phi})}$.

$\square$

## B  Proof of Lemma 2

*Proof.* Since we are allowed to take a singleton decomposition $\mathcal{W}^{(k)} = \mathcal{W}$ and $\mathcal{W}^{(k')} = 0 \ (k' \neq k)$, we have

$$
\begin{aligned}
\left\|\mathcal{W}\right\|_{\overline{S_1/1}} &= \inf_{(\mathcal{W}^{(1)} + \cdots + \mathcal{W}^{(K)}) = \mathcal{W}} \sum_{k=1}^K \|\boldsymbol{W}_{(k)}^{(k)}\|_{S_1} \\
&\leq \|\boldsymbol{W}_{(k)}\|_{S_1} \\
&\leq \sqrt{\underline{r}_k} \|\boldsymbol{W}_{(k)}\|_F \quad (\forall k = 1, \ldots, K)
\end{aligned}
$$

Choosing $k$ that minimizes the right hand side, we obtain our claim.

$\square$

## C  Proof of Theorem 1

*Proof.* Due to the triangular inequality

$$\left\|\hat{\mathcal{W}} - \mathcal{W}^*\right\|_F \leq \left\|\hat{\mathcal{W}} - \mathcal{Y}\right\|_F + \left\|\mathcal{E}\right\|_F.$$

First, due the optimality of $\hat{\mathcal{W}}$, $\mathcal{Y} - \hat{\mathcal{W}} \in \lambda \partial \left\|\hat{\mathcal{W}}\right\|_{\overline{S_1/1}}$, where $\partial\left\|\hat{\mathcal{W}}\right\|_{\overline{S_1/1}}$ is the subdifferential of the latent $S_1/1$ norm at $\hat{\mathcal{W}}$. Thus, the first term of the right hand side satisfies

$$\left\|\hat{\mathcal{W}} - \mathcal{Y}\right\|_F \leq \sqrt{\min_k n_k}\left\|\hat{\mathcal{W}} - \mathcal{Y}\right\|_{\underline{S_\infty/\infty}} \leq \sqrt{\min_k n_k}\lambda,$$

The first inequality is true because $\left\|\mathcal{W}\right\|_F \leq \sqrt{n_k}\|\boldsymbol{W}_{(k)}\|_{S_\infty}$ for any tensor $\mathcal{W}$ and mode $k$. The last inequality is true because for any $\mathcal{G} \in \partial\left\|\hat{\mathcal{W}}\right\|_{\overline{S_1/1}}$, we have $\left\|\mathcal{G}\right\|_{\underline{S_\infty/\infty}} \leq 1$.

Similarly,

$$\left\|\mathcal{E}\right\|_F \leq \sqrt{\min_k n_k}\left\|\mathcal{E}\right\|_{\underline{S_\infty/\infty}} \leq \sqrt{\min_k n_k}\lambda.$$

The last inequality follows from the assumption. This completes the proof of Theorem 1.  $\square$

## D  Proof of Theorem 2

Let $\hat{\mathcal{W}} = \sum_{k=1}^K \hat{\mathcal{W}}^{(k)}$ be the solution and its optimal decomposition of the minimization problem (11); in addition let $\Delta^{(k)} := \hat{\mathcal{W}}^{(k)} - \mathcal{W}^{*(k)}$.

The proof is based on Lemmas 4 and 5, which we present below.

In order to present the first lemma, we need the following definitions. Let $\boldsymbol{U}_k\boldsymbol{S}_k\boldsymbol{V}_k = \boldsymbol{W}_{(k)}^{*(k)}$ be the singular value decomposition of the mode-$k$ unfolding of the $k$th component of the unknown tensor $\mathcal{W}^*$. We define the orthogonal projection of $\Delta^{(k)}$ as follows:

$$\boldsymbol{\Delta}_{(k)}^{(k)} = \boldsymbol{\Delta}_k' + \boldsymbol{\Delta}_k'',$$

where

$$\boldsymbol{\Delta}_k'' = (\boldsymbol{I}_{n_k} - \boldsymbol{U}_k\boldsymbol{U}_k^\top)\boldsymbol{\Delta}_{(k)}^{(k)}(\boldsymbol{I}_{N/n_k} - \boldsymbol{V}_k\boldsymbol{V}_k^\top).$$

Intuitively speaking, $\boldsymbol{\Delta}_k''$ lies in a subspace completely orthogonal to the unfolding of the $k$th component $\boldsymbol{W}_{(k)}^{*(k)}$, whereas $\boldsymbol{\Delta}_k'$ lies in a partially correlated subspace.

The following lemma is similar to Negahban et al. [20, Lemma 1] and Tomioka et al. [26, Lemma 2], and it bounds the Schatten 1-norm of the orthogonal part $\boldsymbol{\Delta}_k''$ with that of the partially correlated part $\boldsymbol{\Delta}_k'$ and also bounds the rank of $\boldsymbol{\Delta}_k'$ .

**Lemma 4.** *Let $\hat{\mathcal{W}}$ be the solution of the minimization problem* (11) *with the regularization constant* $\lambda \geq 2\left\|\mathcal{E}\right\|_{\underline{S_\infty/\infty}}$. *Let $\Delta^{(k)}$ and its decomposition be as defined above. Then we have*

*1.* $\mathrm{rank}(\boldsymbol{\Delta}_k') \leq 2\bar{r}_k.$

*2.* $\sum_{k=1}^K \|\boldsymbol{\Delta}_k''\|_{S_1} \leq 3\sum_{k=1}^K \|\boldsymbol{\Delta}_k'\|_{S_1}.$

Note that although the proof of the above statement closely follows that of Tomioka et al. [26, Lemma 2], the notion of rank is different. In their result, the rank is the mode-$k$ rank $\underline{r}_k$, whereas the rank here is the mode-$k$ rank of the $k$th component $\mathcal{W}^{*(k)}$ of the truth.

The following lemma relates the squared Frobenius norm of the difference of the sums $\left\|\sum_{k=1}^K \Delta^{(k)}\right\|_F^2$ with the sum of squared differences $\sum_{k=1}^K \left\|\Delta^{(k)}\right\|_F^2$

**Lemma 5.** *Let $\hat{\mathcal{W}}$ be the solution of the minimization problem* (11). *Then we have,*

$$\frac{1}{2}\sum_{k=1}^{K}\left\|\Delta^{(k)}\right\|_{F}^{2} \leq \frac{1}{2}\|\Delta\|_{F}^{2} + \alpha(K-1)\sum_{k=1}^{K}\|\mathbf{\Delta}_{(k)}^{(k)}\|_{S_1},$$

*where $\Delta = \sum_{k=1}^{K}\Delta^{(k)}$.*

*Proof of Theorem 2.* First from the optimality of $\hat{\mathcal{W}}$, we have

$$\frac{1}{2}\|\mathcal{Y}-\hat{\mathcal{W}}\|_{F}^{2} + \lambda\sum_{k=1}^{K}\|\hat{\mathbf{W}}_{(k)}^{(k)}\|_{S_1} \leq \frac{1}{2}\|\mathcal{Y}-\mathcal{W}^{*}\|_{F}^{2} + \lambda\sum_{k=1}^{K}\|\mathbf{W}_{(k)}^{*(k)}\|_{S_1},$$

which implies

$$\frac{1}{2}\|\Delta\|_{F}^{2} \leq \langle\Delta,\mathcal{E}\rangle + \lambda\sum_{k=1}^{K}\left(\|\mathbf{W}_{(k)}^{*(k)}\|_{S_1} - \|\hat{\mathbf{W}}_{(k)}^{(k)}\|_{S_1}\right) \tag{21}$$

$$\leq \langle\Delta,\mathcal{E}\rangle + \lambda\sum_{k=1}^{K}\|\mathbf{\Delta}_{(k)}^{(k)}\|_{S_1}$$

$$\leq (\|\mathcal{E}\|_{S_\infty/\infty} + \lambda)\sum_{k=1}^{K}\|\mathbf{\Delta}_{(k)}^{(k)}\|_{S_1}, \tag{22}$$

where we used the fact that $\mathcal{Y} = \mathcal{W}^{*} + \mathcal{E}$ in the first line, the triangular inequality in the second line, and Hölder's inequality in the third line. Note that there is an additional looseness in the third line due to the fact that $\Delta = \sum_{k=1}^{K}\Delta^{(k)}$ is not an optimal decomposition of $\Delta$ induced by the latent Schatten 1-norm.

Next, combining inequality (22) with Lemma 5, we have

$$\frac{1}{2}\sum_{k=1}^{K}\left\|\Delta^{(k)}\right\|_{F}^{2} \leq 2\lambda\sum_{k=1}^{K}\|\mathbf{\Delta}_{(k)}^{(k)}\|_{S_1},$$

where we used the fact that $\lambda \geq \|\mathcal{E}\|_{S_\infty/\infty} + \alpha(K-1)$.

Finally, combining the above inequality with Lemma 4, we have

$$\frac{1}{2}\sum_{k=1}^{K}\left\|\Delta^{(k)}\right\|_{F}^{2} \leq 2\lambda\sum_{k=1}^{K}(\|\mathbf{\Delta}'_k\|_{S_1} + \|\mathbf{\Delta}''_k\|_{S_1})$$

$$\leq 8\lambda\sum_{k=1}^{K}\|\mathbf{\Delta}'_k\|_{S_1}$$

$$\leq 8\lambda\sum_{k=1}^{K}\sqrt{2\bar{r}_k}\|\mathbf{\Delta}'_k\|_{F}$$

$$\leq 8\lambda\sum_{k=1}^{K}\sqrt{2\bar{r}_k}\left\|\Delta^{(k)}\right\|_{F} \tag{23}$$

$$\leq 8\sqrt{2}\lambda\sqrt{\sum_{k=1}^{K}\bar{r}_k}\sqrt{\sum_{k=1}^{K}\left\|\Delta^{(k)}\right\|_{F}^{2}}, \tag{24}$$

where we used the triangular inequality in the first line, Lemma 4 in the second line, Hölder's inequality in the third line (combined with Lemma 4), the fact that $\mathbf{\Delta}_{(k)}^{(k)} = \mathbf{\Delta}'_k + \mathbf{\Delta}''_k$ is an orthogonal decomposition in the fourth line, and Cauchy-Schwarz inequality in the last line. Dividing both sides of the last inequality by $\sqrt{\sum_{k=1}^{K}\|\mathcal{W}^{(k)}\|_{F}^{2}}$ completes the first part of our claim.

In order to obtain the bound on the overall error $\|\Delta\|_{F}$, we consider two cases: if

$$\|\Delta\|_{F}^{2} \leq \sum_{k=1}^{K}\left\|\Delta^{(k)}\right\|_{F}^{2}, \tag{25}$$

we can lower-bound the left hand side of inequality (23) by (25) and obtain

$$\frac{1}{2}\|\Delta\|_{F}^{2} \leq 8\sqrt{2}\lambda\sqrt{\sum_{k=1}^{K}\bar{r}_k}\|\Delta\|_{F}.$$

On the other hand, if

$$\sum_{k=1}^{K} \left\|\!\left\|\Delta^{(k)}\right\|\!\right\|_F^2 \leq \left\|\!\left\|\Delta\right\|\!\right\|_F^2, \tag{26}$$

we can apply the derivation up to (24) to the right hand side of inequality (22) to obtain

$$\frac{1}{2}\left\|\!\left\|\Delta\right\|\!\right\|_F^2 \leq 8\sqrt{2}\lambda \sqrt{\sum_{k=1}^{K} \bar{r}_k} \sqrt{\sum_{k=1}^{K} \left\|\!\left\|\Delta^{(k)}\right\|\!\right\|_F^2}$$

$$\leq 8\sqrt{2}\lambda \sqrt{\sum_{k=1}^{K} \bar{r}_k} \left\|\!\left\|\Delta\right\|\!\right\|_F,$$

where we used assumption (26) in the second line.

We obtain our claim by dividing both sides by $\left\|\!\left\|\Delta\right\|\!\right\|_F$ and choosing $\mathcal{W}^{*(k)} = \mathcal{W}$ for $k = \mathrm{argmin}_k \underline{r}_k$ and $\mathcal{W}^{*(k)} = 0$, otherwise.

$\square$

## E  Proof of Theorem 3

*Proof.* Since each entry of $\mathcal{E}$ is an independent zero men Gaussian random variable with variance $\sigma^2$, for each mode $k$ we have the following tail bound (Corollary 5.35 in [28])

$$P\left(\|\boldsymbol{E}_{(k)}\|_{S_\infty} > \sigma\left(\sqrt{N/n_k} + \sqrt{n_k}\right) + t\right) \leq \exp\left(-t^2/(2\sigma^2)\right).$$

Next, taking a union bound

$$P\left(\max_k \|\boldsymbol{E}_{(k)}\|_{S_\infty} > \sigma \max_k \left(\sqrt{N/n_k} + \sqrt{n_k}\right) + t\right) \leq K \exp\left(-t^2/(2\sigma^2)\right).$$

Thus defining $\delta = K\exp(-t^2/(2\sigma^2))$, we have

$$\left\|\!\left\|\mathcal{E}\right\|\!\right\|_{\underline{S_\infty/\infty}} \leq \sigma \max_k \left(\sqrt{N/n_k} + \sqrt{n_k}\right) + \sigma\sqrt{2\log(K/\delta)} \quad \text{with probability at least } 1 - \delta.$$

Therefore if we take

$$\lambda = 2\sigma\left(\sqrt{N/n_K} + \sqrt{n_1} + \sqrt{2\log(K/\delta)}\right) + \alpha(K-1),$$

the condition of Theorem 2 will be satisfied with probability at least $1 - \delta$. Substituting the above $\lambda$ into the right hand side of the error bound (12) in Theorem 2, we have the statement of Theorem 3.

$\square$

## F  Discussion on the identifiability

Let $\bar{r}_k = \mathrm{rank}(\boldsymbol{W}_{(k)}^{(k)})$ be the mode-$k$ rank of the $k$th component $\mathcal{W}^{(k)}$ in the decomposition

$$\mathcal{W} = \mathcal{W}^{(1)} + \mathcal{W}^{(2)} + \cdots + \mathcal{W}^{(K)}. \tag{27}$$

We say that a decomposition (27) is *locally identifiable* when there is no other decomposition $\sum_{k=1}^{K} \tilde{\mathcal{W}}^{(k)}$ having the same rank $(\bar{r}_1, \ldots, \bar{r}_K)$. The following theorem fully characterizes the local identifiability of the decomposition (27).

**Theorem 4.** *The decomposition* (27) *is* locally identifiable *if and only if* $\mathcal{W}^{(k^*)} = \mathcal{W}$ *for* $k = k^*$ *and* $\mathcal{W}^{(k)} = 0$ *otherwise, for some* $k^*$.

*Proof.* We first prove the "if" direction. suppose that there is another decomposition

$$\sum_{k=1}^{K} \mathcal{W}^{(k)} = \sum_{k=1}^{K} \tilde{\mathcal{W}}^{(k)},$$

such that $\text{rank}(\boldsymbol{W}_{(k)}^{(k)}) = \text{rank}(\tilde{\boldsymbol{W}}_{(k)}^{(k)})$. Note that $\mathcal{W} \neq \tilde{\mathcal{W}}$ can happen only when $\mathcal{W}^{(k)} \neq 0$ (otherwise the rank would increase). Also note that $\mathcal{W} \neq \tilde{\mathcal{W}}$ should happen for at least two $k$'s. Combining these we conclude that there are $k \neq \ell$ such that $\mathcal{W}^{(k)} \neq 0$ and $\mathcal{W}^{(\ell)} \neq 0$.

Conversely, suppose that there are $k \neq \ell$ such that $\mathcal{W}^{(k)} \neq 0$ and $\mathcal{W}^{(\ell)} \neq 0$, we can write[3]

$$\mathcal{W}^{(k)} = \mathcal{C}^{(k)} \times_k \boldsymbol{U}_k,$$
$$\mathcal{W}^{(\ell)} = \mathcal{C}^{(\ell)} \times_\ell \boldsymbol{U}_\ell,$$

where $\boldsymbol{U}_k \in \mathbb{R}^{n_k \times \bar{r}_k}, \mathcal{C}^{(k)} \in \mathbb{R}^{n_1 \times \cdots \times n_{k-1} \times \bar{r}_k \times \cdots \times n_K}$, and $\boldsymbol{U}_\ell$ and $\mathcal{C}^{(\ell)}$ are defined similarly. Since $\mathcal{C}^{(k)}$ and $\mathcal{C}^{(\ell)}$ are allowed to be full rank, we can define

$$\tilde{\mathcal{C}}^{(k)} = \mathcal{C}^{(k)} + \mathcal{D}^{(k,\ell)} \times_\ell \boldsymbol{U}_\ell,$$
$$\tilde{\mathcal{C}}^{(\ell)} = \mathcal{C}^{(\ell)} - \mathcal{D}^{(k,\ell)} \times_k \boldsymbol{U}_k,$$

for any $\mathcal{D} \in \mathbb{R}^{n_1 \times \cdots \times \bar{r}_k \times \cdots \times \bar{r}_\ell \times \cdots \times n_K}$. Then we have

$$\begin{aligned}
\mathcal{W}^{(k)} + \mathcal{W}^{(\ell)} &= \mathcal{C}^{(k)} \times_k \boldsymbol{U}_k + \mathcal{C}^{(\ell)} \times_\ell \boldsymbol{U}_\ell \\
&= \left( \mathcal{C}^{(k)} + \mathcal{D}^{(k,\ell)} \times_\ell \boldsymbol{U}_\ell \right) \times_k \boldsymbol{U}_k \\
&\quad + \left( \mathcal{C}^{(\ell)} - \mathcal{D}^{(k,\ell)} \times_k \boldsymbol{U}_k \right) \times_\ell \boldsymbol{U}_\ell \\
&= \tilde{\mathcal{C}}^{(k)} \times_k \boldsymbol{U}_k + \tilde{\mathcal{C}}^{(\ell)} \times_\ell \boldsymbol{U}_\ell \\
&= \tilde{\mathcal{W}}^{(k)} + \tilde{\mathcal{W}}^{(\ell)}.
\end{aligned}$$

Note that $\text{rank}(\tilde{\boldsymbol{W}}_{(k')}^{(k')}) = \bar{r}_{k'}$ for $k' = k, \ell$. Therefore, there are infinitely many decompositions that have the same rank $(\bar{r}_1, \ldots, \bar{r}_K)$.

$\square$

The above theorem partly explains the difficulty of estimating individual components $\mathcal{W}^{*(k)}$ *without additional incoherence assumption* as in (10). In fact, most decompositions of the form (9) are not identifiable.

# G    Derivation of inequality (13)

*Proof.* Using the triangular inequality and Cauchy-Schwarz inequality we have

$$\left\| \hat{\mathcal{W}} - \mathcal{W}^* \right\|_F \leq \sum_{k=1}^{K} \left\| \hat{\mathcal{W}}^{(k)} - \mathcal{W}^{*(k)} \right\|_F$$
$$\leq \sqrt{K} \sqrt{ \sum_{k=1}^{K} \left\| \hat{\mathcal{W}}^{(k)} - \mathcal{W}^{*(k)} \right\|_F^2 }.$$

Thus

$$\left\| \hat{\mathcal{W}} - \mathcal{W}^* \right\|_F^2 \leq cK\lambda^2 \sum_{k=1}^{K} \bar{r}_k.$$

Now since the decomposition of the true tensor $\mathcal{W}^*$ into the sum $\mathcal{W}^* = \sum_{k=1}^{K} \mathcal{W}^{*(k)}$ is arbitrary, we choose

$$\mathcal{W}^{*(k)} = \begin{cases} \mathcal{W}^* & (\text{if } k = \text{argmin}_k \, \underline{r}_k), \\ 0 & (\text{otherwise}). \end{cases}$$

Therefore $\sum_{k=1}^{K} \bar{r}_k = \min_k \underline{r}_k$, because

$$\bar{r}_k = \begin{cases} \underline{r}_k & (\text{if } k = \operatorname{argmin}_k \underline{r}_k), \\ 0 & (\text{otherwise}), \end{cases}$$

which completes the proof. $\qquad\square$

## H   Proof of Lemma 4

*Proof.* The first inequality is true, because

$$\begin{aligned} \boldsymbol{\Delta}'_k &= \boldsymbol{\Delta}_{(k)}^{(k)} - (\boldsymbol{I}_{n_k} - \boldsymbol{U}_k{\boldsymbol{U}_k}^\top)\boldsymbol{\Delta}_{(k)}^{(k)}(\boldsymbol{I}_{N/n_k} - \boldsymbol{V}_k{\boldsymbol{V}_k}^\top) \\ &= \boldsymbol{U}_k{\boldsymbol{U}_k}^\top\boldsymbol{\Delta}_{(k)}^{(k)}(\boldsymbol{I}_{N/n_k} - \boldsymbol{V}_k{\boldsymbol{V}_k}^\top) + \boldsymbol{\Delta}_{(k)}^{(k)}\boldsymbol{V}_k{\boldsymbol{V}_k}^\top. \end{aligned}$$

Next, to show the second inequality, notice that

$$\begin{aligned} \|\hat{\boldsymbol{W}}_{(k)}^{(k)}\|_{S_1} &= \|\boldsymbol{W}_{(k)}^{*(k)} + \boldsymbol{\Delta}''_k + \boldsymbol{\Delta}'_k\|_{S_1} \\ &\geq \|\boldsymbol{W}_{(k)}^{*(k)} + \boldsymbol{\Delta}''_k\|_{S_1} - \|\boldsymbol{\Delta}'_k\|_{S_1} \\ &= \|\boldsymbol{W}_{(k)}^{*(k)}\|_{S_1} + \|\boldsymbol{\Delta}''_k\|_{S_1} - \|\boldsymbol{\Delta}'_k\|_{S_1} \end{aligned}$$

where we used the decomposability [20] of the Schatten 1-norm in the third line.

Substituting the above inequality into inequality (21), we obtain

$$\begin{aligned} 0 \leq \frac{1}{2}\|\!|\Delta|\!\|_F^2 &\leq \frac{\lambda}{2}\sum_{k=1}^{K}\|\boldsymbol{\Delta}_{(k)}^{(k)}\| + \lambda\sum_{k=1}^{K}(\|\boldsymbol{\Delta}'_k\|_{S_1} - \|\boldsymbol{\Delta}''_k\|_{S_1}) \\ &\leq \lambda\sum_{k=1}^{K}\left(\frac{3}{2}\|\boldsymbol{\Delta}'_k\|_{S_1} - \frac{1}{2}\|\boldsymbol{\Delta}''_k\|_{S_1}\right), \end{aligned}$$

from which the statement follows. Here we used $\|\!|\mathcal{E}|\!\|_{\underline{S_\infty/\infty}} \leq \lambda/2$ in the second inequality and the triangular inequality in the last line. $\qquad\square$

## I   Proof of Lemma 5

*Proof.*

$$\begin{aligned} \|\!|\Delta|\!\|_F^2 &= \sum_{k=1}^{K}\|\!|\Delta^{(k)}|\!\|_F^2 + \sum_{k=1}^{K}\sum_{k'\neq k}\left\langle \boldsymbol{\Delta}_{(k)}^{(k)}, \boldsymbol{\Delta}_{(k)}^{(k')} \right\rangle \\ &\geq \sum_{k=1}^{K}\|\!|\Delta^{(k)}|\!\|_F^2 - \sum_{k=1}^{K}\|\boldsymbol{\Delta}_{(k)}^{(k)}\|_{S_1}\sum_{k'\neq k}\|\boldsymbol{\Delta}_{(k)}^{(k')}\|_{S_\infty} \\ &\geq \sum_{k=1}^{K}\|\!|\Delta^{(k)}|\!\|_F^2 - 2\alpha(K-1)\sum_{k=1}^{K}\|\boldsymbol{\Delta}_{(k)}^{(k)}\|_{S_1}, \end{aligned}$$

where the last inequality follows from $\|\boldsymbol{\Delta}_{(k)}^{(k')}\|_{S_\infty} \leq \|\hat{\mathcal{W}}_{(k)}^{(k')}\|_{S_\infty} + \|\mathcal{W}_{(k)}^{*(k')}\|_{S_\infty} \leq 2\alpha$ for $k' \neq k$. Lemma follows by dividing both sides by two. $\qquad\square$

## Footnotes

[3] Here the tensor mode-$k$ product $\mathcal{A} = \mathcal{B} \times_k \boldsymbol{C}$ is defined as $a_{i_1 \ldots i_K} = \sum_{\ell=1}^{d_k} b_{i_1 i_2 \ldots \ell \ldots i_K} c_{\ell i_k}$ where $\mathcal{A} = (a_{i_1 \ldots i_K}) \in \mathbb{R}^{n_1 \times \cdots \times n_K}, \mathcal{B} = (b_{i_1 \ldots \ell \ldots i_K}) \in \mathbb{R}^{n_1 \times \cdots \times d_k \times \cdots \times n_K}$, and $\boldsymbol{C} = (c_{\ell i_k}) \in \mathbb{R}^{d_k \times n_k}$