[Reviews · NeurIPS 2013]

Submitted by Assigned_Reviewer_1

This paper considers convex norm-regularized optimization approaches for factorization of approximately low-rank tensors. It focuses on a recently proposed "latent" norm approach ([25]), and its comparison to a previously used "overlap" norm approach. Specifically, the paper derives theoretical bounds on the "latent" norm mean-squared-error behavior under several noise assumptions, which compare favorably to the existing analysis on the overlap norm, supporting its better empirically-observed performance in a previous paper. The theory is also confirmed via some numerical simulations. Finally, the paper considers a generalization of both these norms, and proves a duality theorem for it.

The paper is generally easy to follow, and appears to be technically sound (although I haven't checked the proofs in the supplementary material carefully). The results may also be useful for researchers focusing on optimization methods for tensor decompositions. My main issues are that
(i) The results are narrow and of a technical nature, only peripherally related to machine learning, and probably of limited potential impact at NIPS. While the authors make the (justified) case that tensor factorizations can crop up in machine learning, the paper does not attempt to examine the proposed approaches in the context of machine learning problems, either theoretically or experimentally.
(ii) The results seem a bit incremental, and I didn't find them to be particularly surprising (admittedly this is a subjective notion). They seem to provide some additional theoretical backing to a specific recently proposed approach [25], which was already argued empirically to work well. The techniques used also don't seem particularly novel.

A more minor defect is that the upper bounds do not come with any lower bounds. This is potentially dangerous, since the upper bounds are used to compare between different algorithms and argue for the superiority of one over the other.

Additional Comments
-----------------

- The abstract begins with "We propose a new class of structured Schatten norms for tensors...". However, I think this is slightly misleading. The focus on the paper is devoted to analyzing two existing approaches (overlapped and latent). Unless I missed something, the generalization only comes up by defining it in section 2.1 and proving what is the dual norm (Lemma 1). In particular, no particular algorithms are proposed based on this generalized Schatten norms,
the behavior of such algorithms is not analyzed, nor are they tested experimentally.

- Figure 1 is presented in the introduction, but at this stage is not interpretable without the background appearing in section 2 onwards.

- Line 38: "Tensor is" -> "Tensor modelling is"(?)
- Line 40: Missing comma after "Conventionally"
- Line 395: "also in such situations" -> "in such situations as well"
Summary: The paper's main contribution are theoretical error bounds for a recently proposed low-rank tensor decomposition approach. The paper seems technically sound, but the results are somewhat incremental and may suffer from limited impact at NIPS.

Submitted by Assigned_Reviewer_8

****Additional comments after the rebuttal:
The authors need to do a much better job on motivating the problem for machine learning, for the paper to be accessible to a wider audience. The authors need to provide the latent variable model (with a graphical model representation) where this tensor decomposition can learn the parameters: Consider a model where latent variable h_1 connects to observed variable x_1, h_2 to x_2 and so on. Assume a linear model from hidden to observed variables, and arbitrary dependence among the h_i's. In this case, the observed moment tensor has a tucker decomposition and its multilinear rank is dim(h_1), dim(h_2).. The authors need to add this discussion and preferably provide a picture depicting this model. The authors also need to point out that it is possible that only some hidden variables are of small dimensions and not the others, which was a main advantage for the new result in the paper.

The paper considers convex-optimization based tensor decomposition using structured Schatten norms for tensors. The authors employ the overlapped Schatten norm and show how it leads to better reconstruction guarantees. The paper is well written and makes technically important contributions. Fixing a few technical and notational issues, will make the paper even more readable.


*Clarification about Theorem 2 and its proof:

Statement of theorem 2 is technically not correct: the components are only recovered up to a permutation. Although a minor issue, the authors need to mention this.

In Theorem 2, it is stated that (12) immediately leads to (13). This needs a justification.

Proof of Lemma 5 is missing. Lemma 5 forms a key part of Theorem 2.

It is better to clarify the proof of Lemma 4 explicitly rather than referring to another paper, although it is straightforward.

*Minor issues:

Need to add more related work on HOSVD and other methods for tensor decomposition:
The authors can mention how the higher order SVD and its variants yields approximation bounds for tensor decomposition. See Kolda's review on tensor decompositions (section 4).

The notion of rank that the authors use, which they refer to as mode-rank is also referred ot as multilinear rank. The authors can mention this and note there are different notions of tensor rank in the introduction.

The authors can briefly discuss the computational complexity of the methods employed.

The authors need to clarify their discussion about incoherence after (10). The authors use the term to mean that they need the "energy" to be spread across different mixture components and modes, i.e., no component or mode can have too high a singular value. The authors mention that this is the same as the incoherence conditions of [1,3,10]. I do not see this. In [1,3,10], incoherence notions are use for sparse+low rank recovery. However, here, the noise is not sparse. The condition of [1,3,10] involves the singular vectors being incoherent with the basis vectors. This is different from the condition here. Suggest removing this comparison.

*Typos/notational issues:

Superscript notation for components \hat{\cal{W}}^{(k)} is confusing. Suggest using some other symbol for components.

Bad notation: \bar{r}_k and \underline{r}_k are both used for k-mode rank. This should be changed. Could introduce mult_rank(tensor) instead to denote the multilinear rank.

In Last line of page 12 in supplementary material, some parenthesis is missing.

Abstract: "result for structures Schatten" should change to structured
Summary: The paper is well written and makes novel contribution in terms of finding a good Tucker tensor decomposition using an overlapped Schatten norm and provides guarantees for recovery. There are however some technical clarifications needed which will hopefully be addressed by the authors.

Submitted by Assigned_Reviewer_9

This paper provides theoretical bounds on the performance of a convex tensor decomposition algorithm regularized by the latent Schatten norm. It improves upon the result known for regularization using the overlapped Schatten norm.

Quality : The paper obtains theoretical results for tensor decomposition using latent Schatten norm. The dual characterization of the overlapped and latent Schatten norm in particular is interesting. The experiments however are rather briefly considered and are not elaborated upon. For instance how were the Tucker ranks (r_1,...r_k) of the examples chosen? How does group regularization such as l_1,\infty work when the tensor is simultaneously low rank in all modes? How does the method perform as compared to non-convex approaches?

Clarity : The paper is well organized and clear to understand.

Originality : The main original result in the paper is the duality between the overlapped and latent Schatten norms. This result seems to be of independent interest.

Significance : Tensor decomposition is an important problem encountered in many applications. Since this paper improves upon existing results for this problem, I believe it is significant work.
Summary: This paper improves upon existing results for the tensor decomposition problem. The theoretical results are interesting and the paper is well written. However, experiments are considered rather briefly towards the end of the paper.
Author Feedback

Author rebuttal: We would like to thank all reviewers for their helpful comments.

To Assigned_Reviewer_1:
It is still an open question whether there is a (class of) computationally tractable convex relaxation for tensor rank, and this work, though it may seem incremental, is a step toward this goal. A convex relaxation for tensor decomposition is relevant not only in machine learning and data mining, but also for quantum physics and computational complexity theory. Thus we believe that it is suitable for an interdisciplinary venue such as NIPS.

A lower bound for tensor decomposition was recently discussed in Mu et al. (2013) "Square Deal: Lower Bounds and Improved Relaxations for Tensor Recovery". Although their result does not directly apply to our case, it is likely that a similar argument could show that our bound is tight, which is also consistent with our numerical results.

"We propose a new class of structured Schatten norms for tensors..." refers to the fact that we propose overlapped/latent Schatten S_p/q-norms, which is more general than the p=1 and q=1 case studied earlier. The dual norm (p=\infty and q=\infty) plays a crucial role for our analysis. Moreover, other choices (e.g., p=1 and q=\infy) may also be used in the future.

Other suggestions are reflected in the latest version of our manuscript.

To Assigned_Reviewer_8:
Statement of Theorem 2 is correct because each component is regularized to be low rank along its own mode. Since being low-rank (along a specific mode) is not an invariant property, there is no ambiguity of permutation. Nevertheless we need the incoherence assumption (10) since if a component happens to be simultaneously low-rank along different modes, it makes the decomposition ambiguous.

The incoherence assumption is not specific to the sparse+low-rank decomposition. In both our model and the sparse+low-rank decomposition, the signal is decomposed into different parts that obey different types of sparsity or low-rank-ness. In fact, the assumption (10) controls the degree of redundancy; a simultaneously sparse and low-rank component is bad for sparse+low-rank decomposition, and a component that is simultaneously low-rank along different modes is bad for our model.

We have added the derivation of (13) and proofs for Lemmas 4 and 5 in the appendix.

We will add more references to HOSVD and related methods.

We agree that multilinear rank is a more common terminology. This will be fixed in the final version.

Regarding the computational complexity, all the operations except SVD requires O(n^K) (this is unavoidable because all entries are observed). The cost of truncated SVD is O(rn^K) using Mark Tygert's randomized SVD algorithm. The number of iterations is usually on the order of hundreds and does not depend on the size of the problem.

\bar{r}_k refers to the mode-k rank of the k-th component W^{(k)} and \underline{r}_k refers to the mode-k rank of the sum \sum_{k}W^{(k)}; thus they are different.

Other notational suggestions will be reflected in the final version.

To Assigned_Reviewer_9:
In the experiments, the mode-k rank was chosen uniformly at random from [1,...,n_k] for k=1,...,K. We geneerated 20 random instances for each tensor size (50x50x20 or 80x80x40). We added Gaussian noise to these instances and averaged the estimation error (over 10 random realization of the noise) for each instance, which corresponds to a single data point in Figure 2. This will be explained more clearly in the final version.

Group regularization would be an interesting future direction.